# Parvoviruses at the Heart: Endothelial Injury and Myocyte Lysis in Human B19V and Canine CPV-2 Infections

**DOI:** 10.3390/cimb48010052

**Published:** 2025-12-31

**Authors:** Anna Golke, Maciej Przybylski, Wojciech Mądry, Michał Buczyński, Agata Moroz-Fik, Tomasz Dzieciątkowski, Tadeusz Frymus, Olga Szaluś-Jordanow

**Affiliations:** 1Department of Preclinical Sciences, Institute of Veterinary Medicine, Warsaw University of Life Sciences-SGGW, Ciszewskiego 8, 02-786 Warsaw, Poland; 2Chair and Department of Medical Microbiology, Medical University of Warsaw, Chałubińskiego 5, 02-004 Warsaw, Poland; 3Department of Heart, Chest and Transplant Surgery, Medical University of Warsaw, Żwirki i Wigury 63A, 02-091 Warsaw, Poland; 4Division of Veterinary Epidemiology and Economics, Institute of Veterinary Medicine, Warsaw University of Life Sciences-SGGW, Nowoursynowska Str. 159c, 02-776 Warsaw, Poland; 5Department of Small Animal Diseases with Clinic, Institute of Veterinary Medicine, Warsaw University of Life Sciences-SGGW, Nowoursynowska Str. 159c, 02-776 Warsaw, Poland; tadeusz_frymus@sggw.edu.pl

**Keywords:** parvovirus B19, canine parvovirus type 2, viral myocarditis, dilated cardiomyopathy, endothelial injury, cardiomyocyte lysis, NS1 protein, diagnostic evidence bundles, cardiac biomarkers, troponin

## Abstract

Background: Parvovirus B19 (B19V; *Erythroparvovirus primate 1*) is now the most commonly detected virus in human endomyocardial biopsies from patients with myocarditis or dilated cardiomyopathy; however, its true causal role remains uncertain. By contrast, *Protoparvovirus carnivoran 1*, also known as canine parvovirus type 2 (CPV-2), is an apparent cause of myocarditis in neonatal puppies, where it replicates in cardiomyocytes, induces extensive cell death, and often leaves fibrotic scars in survivors. Conclusions: This review compares B19V and CPV-2 from basic biology to clinical expression. Divergent tropism and replication kinetics produce distinct injury patterns: predominantly endothelial and microvascular dysfunction with immune-mediated damage in adult human B19V infection versus direct, age-restricted cardiomyocyte lysis in neonatal CPV-2 infection, often followed by fibrosis. Because parvoviral DNA can persist in cardiac tissue, detection alone does not prove causality. We advocate an “evidence bundle” integrating viral load by quantitative polymerase chain reaction (qPCR), detection of viral transcripts and/or proteins when feasible, spatial co-localization with histological injury, and concordant clinical markers (cardiac troponins and advanced imaging, including cardiac magnetic resonance imaging [CMR]) to support etiologic attribution and guide management in human and veterinary cardiology.

## 1. Introduction

Viral infections remain one of the principal causes of acute myocarditis and a significant driver of dilated cardiomyopathy, especially in younger patients for whom coronary artery disease is uncommon [1]. Viral myocarditis encompasses a wide clinical spectrum—from fulminant forms with cardiogenic shock, through acute chest-pain syndromes mimicking myocardial infarction, to subacute or chronic forms progressing to dilated cardiomyopathy with ventricular remodelling [1]. Age markedly modifies the phenotype: infants and children frequently present with acute heart failure, arrhythmias, or sudden death, whereas adults more often present with chest pain, troponin elevation, and later with chronic systolic dysfunction [2].

Recently, endomyocardial biopsy and molecular diagnostics have revealed an evolving viral spectrum in human myocarditis. Enteroviruses and adenoviruses, once predominant, have declined, while B19V and human herpesvirus 6 (HHV-6) have emerged as the most frequently detected agents in cardiac tissue [1,3,4].

This shift from enteroviruses and adenoviruses toward frequent detection of B19V in cardiac tissue has renewed interest in parvoviruses as potential drivers of myocardial injury, while also highlighting the challenge of etiologic attribution in humans and the value of comparative insight from CPV-2 myocarditis in dogs.

Since 2023, surveillance across Europe has recorded a marked post-pandemic resurgence of B19V infections, including clusters of myocarditis and fetal hydrops. Epidemiological analyses have shown that COVID-19-associated public health restrictions altered viral circulation, leading to an “immunity gap” and an atypically large rebound of B19V activity in 2024–2025 [5,6,7,8]. These outbreaks were accompanied by increased detection of this virus among pregnant women and by higher rates of anemia and myocarditis in children and young adults [9].

The interplay between B19V and other viruses, particularly dengue virus (DENV) and severe acute respiratory syndrome coronavirus 2 (SARS-CoV-2), has also gained attention. Coinfection studies in Asia have demonstrated that B19V may exacerbate endothelial injury and immune dysregulation in dengue patients, potentially contributing to more severe cardiac or hematologic manifestations [10]. Such observations reinforce the broader recognition that parvoviruses can amplify disease severity through immune and endothelial mechanisms. In this context, B19V has re-emerged as the prototypical endotheliotropic parvovirus associated with myocardial inflammation and dysfunction. Parallel discoveries have expanded the known host range of erythroparvoviruses. A 2025 metagenomic investigation identified a novel parvovirus in domestic cats, closely related to human B19V, provisionally classified within *Erythroparvovirus*. This suggests that parvoviruses capable of endothelial or erythroid tropism may circulate more broadly among mammals than previously recognized [11]. In the future, this feline *Erythroparvovirus* can potentially provide additional translational insights into the biology and cardiac impact of B19V in humans.

In veterinary medicine, CPV-2, remains one of the most reliable natural models of parvoviral myocarditis in a mammalian host. In neonatal puppies, CPV-2 infects proliferating cardiomyocytes, leading to acute necrotizing myocarditis with high mortality, whereas in older dogs, it causes enteritis with secondary subclinical or clinical cardiac involvement. This striking age-restricted susceptibility mirrors, to some extent, the vulnerability of human infants to severe parvoviral myocarditis, including fetal heart failure [12,13].

Nevertheless, the biology, tropism, and clinical expression of B19V and CPV-2 differ profoundly. The B19V targets endothelial and erythroid cells, while CPV-2 infects rapidly dividing myocytes and epithelial cells of the intestinal crypts. Direct extrapolation from canine to human infection is therefore not appropriate. A comparative analysis of their virology, replication mechanisms, host responses, and diagnostic evidence—structured within a multi-layered “evidence-bundle” framework—is essential to clarify how small, autonomous DNA viruses can trigger or modulate myocardial injury across species.

## 2. Biology of Parvoviruses

### 2.1. Parvoviruses as Small Autonomous DNA Viruses

*Parvoviridae*, one of the families of smallest animal viruses, are characterized by non-enveloped icosahedral capsids of approximately 20–26 nm enclosing a linear single-stranded DNA genome of about 5–6 kb [14]. This family is currently divided into multiple genera; human parvovirus B19 (B19V) is placed in the genus *Erythroparvovirus*, whereas canine parvovirus type 2 (CPV-2) is classified within the genus *Protoparvovirus* [15,16]. A fundamental distinction is made between autonomous parvoviruses, which complete their replication cycle without assistance from other viruses, and dependoparvoviruses (such as adeno-associated viruses) that require helper functions, typically provided by adenoviruses or herpesviruses, to achieve productive replication [17]. Both B19V and CPV-2 are autonomous; however, their extremely compact genomes encode only a limited set of non-structural and structural proteins, thereby imposing a strict reliance on the host cell’s S-phase machinery for DNA synthesis and on host regulatory pathways for replication and gene expression [14,17].

In autonomous parvoviruses, the major non-structural protein 1 (NS1) is the central replication and regulatory factor. NS1 contains conserved functional domains that mediate origin-specific DNA nicking (endonuclease activity) and superfamily 3 (SF3) helicase/adenosine triphosphatase (ATPase) activity, and it has been shown to modulate viral and cellular transcription, together coordinating rolling-hairpin replication of the single-stranded DNA (ssDNA) genome [18,19,20,21].

Because parvoviruses do not encode a viral DNA polymerase, viral DNA synthesis relies on host DNA polymerases and other S-phase (DNA synthesis phase) replication factors, creating a strong dependence on cell-cycle progression and a preference for actively proliferating tissues [22]. This reliance underlies characteristic tissue tropisms, including replication in erythroid progenitor cells during B19V infection and in rapidly dividing intestinal crypt epithelium or neonatal cardiomyocytes in CPV-2 infection [12,23,24,25,26].

### 2.2. Virion Structure of B19V and CPV-2

Parvovirus particles are non-enveloped triangulation number 1 (T = 1) icosahedral capsids of ~23–26 nanometers (nm) diameter, assembled from 60 capsid subunits. For B19V, this architecture has been resolved by cryogenic electron microscopy (cryo-EM), including atomic-resolution structures of authentic virions purified from patient blood [27,28,29].

In B19V, the capsid is built predominantly from viral protein 2 (VP2) with a small number of viral protein 1 (VP1) subunits (VP1 contains an N-terminal VP1-unique region–VP1u) [23]. VP2 accounts for about 95% of capsid subunits and forms the canonical eight-stranded β-barrel scaffold and surface loops that create the threefold spikes and twofold depressions, whereas VP1 contributes roughly 5% of subunits and differs by an VP1u region of about 227 amino acids that is flexibly exposed at the capsid surface [29]. The VP1u region carries a conserved phospholipase A2 (PLA2) motif that is essential for endosomal escape and efficient infection, and harbours conformational epitopes that dominate the neutralizing antibody response and participate in receptor engagement and membrane perturbation [30,31]. Functional studies have shown that the PLA2 activity of VP1u and its Ca^2+^-dependent conformational changes are tightly coupled to endosomal membrane penetration and cytotoxicity [30]. Interestingly, patient-derived virions were observed to circulate in complex with the host protease inhibitors—inter-alpha-trypsin inhibitor heavy chain 4 (ITIH4) and serpin family A member 3 (SERPINA3)—bound to the capsid surface [28].

In B19V linear, ssDNA genome of approximately 5.6 kb encodes NS1, the capsid proteins, and several small non-structural proteins, and is flanked by terminal inverted repeats that fold into hairpins, functioning both as self-priming origins of replication and as structural elements in genome packaging [15,32].

For CPV-2, capsid architecture was first established by X-ray crystallography [33] and has been further characterized using complementary approaches, including monoclonal antibody neutralization and escape-mutant mapping of surface loops [34], cryo-EM of transferrin receptor type 1 (TfR1)–bound capsids with biochemical binding analyses [35], and protease-susceptibility assays capturing VP2-to-VP3 processing in genome-containing particles [36]. Together with other studies, these data helped define CPV-2 capsid architecture and capsid–genome features [29]. Structurally, CPV-2 virion architecture is broadly similar to that of B19V, with a T = 1 icosahedral capsid built from 60 copies of VP2 as the major structural component and a minority of VP1 molecules that provide an N-terminal VP1u segment containing a PLA2 motif analogous to that of B19V [29]. However, the amino acid composition and surface topology of VP2 differ substantially between CPV-2 and B19V, reflecting their adaptation to distinct cellular receptors and host species. In CPV-2, residues located in surface loops around the threefold spikes and adjacent depressions form the footprint for canine TfR1; subtle substitutions in these loops alter receptor affinity, host range, and pandemic potential [37,38,39]. Cryo-EM has shown that TfR1 binds asymmetrically to the capsid surface and engages a dynamic interface, providing a structural explanation for species-specific receptor usage and for how relatively few capsid mutations can shift tropism between feline, canine, and other carnivore hosts [35,39]. Moreover, in genome-containing virions, a proportion of VP2 is proteolytically processed to VP3 (N-terminal cleavage of ~19 residues), a feature that is further discussed in the context of parvovirus entry and capsid remodeling [36,40].

In contrast, B19V capsid surface features mediate interaction with globoside (P antigen) and additional erythroid-restricted factors, rather than TfR1, underpinning its primary erythroid and endothelial tropism in humans [23]. Interestingly, receptor-related work has demonstrated that globoside is not absolutely required for virus entry but is critical at a post-entry step for productive infection, implying that additional proteinaceous receptors or co-receptors shape B19V tropism at the level of erythroid progenitors and endothelial cells [41].

### 2.3. Replication Cycles and Cell Tropism

Capsid–receptor interactions are key determinants of cell entry and host range; therefore capsid properties strongly shape tissue tropism and, consequently, typical injury patterns [40,42]. In B19V infection, attachment occurs via high-affinity binding of the capsid to the neutral glycosphingolipid globoside on erythroid progenitor cells in bone marrow and fetal liver, as well as on selected endothelial and placental cells [43,44,45,46]. As mentioned before, globoside is not strictly required for entry but is critical at a post-entry step for productive infection, whereas a distinct VP1u-dependent receptor interaction is essential for internalisation [41,47]. Co-receptors, including α5β1 integrin and Ku80, have been implicated in facilitating B19V entry and post-attachment signalling in non-erythroid cells, particularly endothelial cells [48,49]. Following receptor engagement, human parvovirus B19 (B19V) virions are internalised by clathrin-mediated endocytosis and trafficked through acidified endosomes, where conformational rearrangements expose the VP1 unique region (VP1u). VP1u contains a conserved PLA2 motif, and this enzymatic activity is required for endosomal escape and subsequent delivery of capsids to the nucleus [31,50,51]. Importantly, these VP1u-mediated steps primarily define an entry requirement for infection rather than a tissue-restricted injury program.

After nuclear entry, the ssDNA genome is converted by host polymerases into a double-stranded replicative form, and NS1 then orchestrates rolling-hairpin replication from the terminal repeats by nicking and unwinding DNA to generate progeny strands, consistent with *Parvoviridae* replication strategies [14,21,23]. Mechanistically, NS1 integrates origin recognition with site-specific endonuclease and superfamily 3 (SF3) helicase/adenosine triphosphatase (ATPase) functions, enabling iterative genome amplification [21,52]. Beyond NS1, auxiliary non-structural proteins can measurably increase progeny output: in B19V, the 11-kDa protein augments viral DNA replication and increases released progeny virion output by ~7–10-fold, typically quantified as a fold change in DNase-resistant released genomes and/or infectious units [23]. Accordingly, tissue susceptibility and injury in B19V infection are driven chiefly by where productive replication is supported.

Productive B19V replication is confined mainly to erythroid progenitor cells at the burst-forming unit–erythroid (BFU-E) and colony-forming unit–erythroid (CFU-E) stages, where erythropoietin-driven signalling, erythroid transcription factor expression, and S-phase entry create a permissive environment; in these cells, high-level genome amplification and capsid assembly culminate in lysis and can precipitate transient aplastic crises in susceptible hosts [23,53]. In primary human erythroid progenitor cell models, productive infection yields measurable infectious progeny virus, with titers reported to exceed 90 focus-forming units per microliter (FFU/µL) after passage under hypoxia, compared with <5 FFU/µL under normoxia; reverse-genetics systems under hypoxia have reported titers of ~150 FFU/µL [54].

In contrast, infection of endothelial cells in the adult myocardium is usually restricted or abortive, with limited genome amplification and expression of NS1 and other non-structural proteins, and the replication cycle is typically incomplete with no detectable infectious progeny virions produced in endothelial cells (reported as “restricted transcription/replication” rather than productive virion release) [55]. Nevertheless, this restricted replication is sufficient to trigger pro-inflammatory signaling, apoptosis, and endothelial-to-mesenchymal transition [56,57].

Adult cardiomyocytes, which are largely post-mitotic and do not express P antigen, are considered non-permissive for productive B19V infection, although B19V DNA and, more rarely, capsid proteins have been detected in myocardial tissue in association with lymphocytic myocarditis and dilated cardiomyopathy, findings that are interpreted as evidence of endothelial infection with bystander myocyte damage rather than robust myocyte replication [58]. In fetuses, cardiomyocytes differ because P antigen expression and proliferative activity have been documented, and intrauterine B19V infection has been associated with fetal myocarditis and hydrops fetalis secondary to severe anemia and cardiac dysfunction, indicating that direct cardiomyocyte infection can occur in that developmental stage [44,59]. This compartmentalisation has direct diagnostic implications: in adults, the most informative tissue-based signals are those indicating endothelial localisation and transcriptional activity (rather than low-copy DNA alone), whereas in fetal disease, evidence consistent with cardiomyocyte permissivity may carry greater weight.

CPV-2 follows a broadly similar replication strategy but exploits different receptors and target cells. The CPV-2 capsid binds to the canine TfR1 with high affinity, and specific amino acid substitutions in VP2 that arose during the emergence of a feline panleukopenia virus ancestor have been shown to confer efficient binding and entry into canine cells, defining the host range [38]. After TfR-mediated endocytosis and trafficking through acidified endosomes, exposure of VP1u and its PLA2 activity again enables endosomal escape and nuclear delivery, after which viral DNA replication proceeds in rapidly dividing cells in strict dependence on host S-phase machinery [15,17]. In vivo, productive CPV-2 replication occurs in mitotically active intestinal crypt epithelial cells, bone marrow progenitors, and, critically for myocarditis, cardiomyocytes in fetal and early neonatal puppies that remain in the cell cycle after birth [13,60].

Permissivity of canine cardiomyocytes to CPV-2 is confined to a narrow age window from late gestation to roughly the first two to three weeks post-partum; beyond this period, withdrawal of cardiomyocytes from the cell cycle renders the myocardium largely refractory to productive infection, and CPV-2 infection in older puppies is dominated by enteritis and bone marrow suppression with only indirect, functional cardiac involvement [60,61]. Accordingly, age serves as a key diagnostic criterion: direct myocardial infection is most plausibly supported by cardiomyocyte-centred lesions with intranuclear inclusions and concordant viral detection in very young pups, whereas similar clinical signs outside this window require more cautious interpretation and stronger tissue-based proof of local replication. The taxonomy, virion/genome architecture, receptor usage, and entry mechanisms discussed above are summarised in Table 1.

## 3. From Myocardial Injury to Clinical Care

### 3.1. Direct Viral Injury and Immune-Mediated Mechanisms in Myocardial Damage

In B19V-associated heart disease, an endothelium-centric pattern of injury has been described. In situ hybridization and immunohistochemical studies have localized B19V genomes and proteins predominantly to endothelial cells of intramural coronary vessels and myocardial capillaries, with only rare detection in cardiomyocytes [58,62,63]. NS1 and the small 11 kDa protein can induce apoptosis, DNA damage, and pro-inflammatory gene expression in human endothelial progenitor and microvascular endothelial cells, including activation of signal transducer and activator of transcription (STAT) and nuclear factor kappa-light-chain-enhancer of activated B cells (NF-κB) pathways and up-regulation of cytokines such as interleukin-6 and monocyte chemoattractant protein-1 [32,56,64]. These data support a model in which low-level or abortive B19V replication in myocardial endothelium drives microvascular dysfunction, endothelial loss, and microinfarction, with vascular injury rather than myocyte infection as the primary viral target [31,65].

In vivo, B19V-positive myocarditis has been associated with reduced coronary flow reserve, patchy perfusion defects, and microvascular rarefaction, accompanied by lymphocytic infiltrates cantered on small vessels and extending into the interstitium—findings consistent with combined direct endothelial injury and immune-mediated bystander damage [2,66]. Increased myocardial expression of interferon-γ (IFN-γ), tumor necrosis factor-α (TNF-α), and other pro-inflammatory mediators has been reported in B19V-positive biopsies, suggesting that cytokine networks further depress contractility and promote apoptosis [1,31,67]. The resulting clinical presentations range from fulminant cardiogenic shock to arrhythmogenic “hot phases” and effusive–constrictive pericarditis, underscoring the contribution of both microvascular injury and dysregulated immune responses to the phenotype [68,69,70,71].

B19V DNA has been found at low copy numbers in endomyocardial biopsies or autopsy samples from individuals with and without cardiomyopathy, as well as in other tissues, indicating that capsid DNA alone is not sufficient to establish causality [58,72,73]. Consequently, the pathogenic relevance of B19V is now judged less by the presence of DNA and more by markers of transcriptional activity and associated inflammation. Therefore, rather higher viral loads, detection of viral mRNA or capsid/NS1 protein, and spatial co-localisation with endothelial damage and lymphocytic infiltrates are considered stronger evidence that B19V is etiologically involved [2,55,74]. This “evidence bundle” approach has also been adopted in therapeutic trials, where modulation of B19V transcriptional activity, rather than DNA clearance alone, correlates with clinical improvement [57].

In CPV-2–associated myocarditis, direct lytic infection of cardiomyocytes with myocardial necrosis—often affecting the left ventricle and interventricular septum—and with intranuclear inclusion bodies in cardiomyocytes and a predominantly lymphoplasmacytic infiltrate has been demonstrated [13,60,75]. CPV-2 DNA and antigen have been found in cardiomyocyte nuclei, and viral mRNA has been detected in affected hearts, confirming active replication at the site of injury [12,76]. CPV-2 NS1 is a highly cytotoxic protein that induces DNA damage, apoptosis, and shutdown of host protein synthesis [19,77].

The inflammatory response contributes to myocardial damage in both infections, but timing and relative weight differ. In neonatal CPV-2 myocarditis, massive viral replication and direct cytolysis appear to account for most of the acute tissue destruction, with the lymphoplasmacytic infiltrate largely representing clearance of necrotic and infected cells [13,60,75]. The healing phase is characterized by rapid replacement fibrosis, resulting in thinned, scarred ventricular walls in survivors and a risk of chronic systolic dysfunction and arrhythmias later in life [12,78]. In puppies older than 2–3 weeks, myocardial injury is usually subclinical and driven mainly by systemic inflammatory and hemodynamic stress rather than by extensive cardiac viral replication. Moreover, increases in cardiac troponin I and transient impairment of myocardial deformation on speckle-tracking echocardiography have been documented, consistent with reversible myocardial involvement without overt necrotizing myocarditis [79,80].

In B19V infection, the immune component is more prominent and often longer-lasting. Lymphocytic myocarditis with scattered myocyte necrosis, fulfilling the Dallas criteria, has been described in B19V-positive biopsies; however, cardiomyocyte damage is thought to be secondary, mainly to microvascular injury and T-cell/macrophage-mediated effects, rather than to productive infection of myocytes [1,2]. Persistent B19V DNA and expression of viral transcripts have been linked in some cohorts to chronic inflammation and progression to post-inflammatory dilated cardiomyopathy, whereas other studies have reported B19V genomes as incidental findings with no clear clinical impact, underlining that host context, viral activity, and the surrounding immune milieu are decisive determinants of pathogenicity [3,63,72]. The cardiac targets, age windows, permissivity drivers, dominant injury patterns, histologic hallmarks, and clinical phenotypes discussed above are summarised in Table 2.

### 3.2. Diagnostic Approaches and Evidence Bundles

Confirmation of suspected viral myocarditis in humans has been shaped by consensus documents that recommend integration of clinical, imaging, and histological data, with cardiovascular magnetic resonance (CMR) and endomyocardial biopsy (EMB) [1,81,82]. CMR, using the updated Lake Louise criteria, can detect myocardial edema, hyperemia, as well as replacement fibrosis and has substantially improved the sensitivity of non-invasive diagnosis of non-ischemic myocarditis [81]. However, attribution of the disease to a specific virus still depends on tissue-based detection of viral nucleic acids and, ideally, viral transcripts or proteins in conjunction with histological lesions [1].

EMB provides direct access to myocardial tissue but is constrained by small sample size, spatial heterogeneity of lesions, and pre-analytical and analytical variability across laboratories [1,74]. For B19V, an “evidence bundle” approach has been recommended to prevent overinterpretation of low-level DNA positivity. In this framework, quantitative PCR for viral DNA with appropriate normalization and genotype coverage, demonstration of transcriptional activity or protein expression by strand-specific RT-qPCR, in situ hybridization, or immunohistochemistry, and spatial co-localisation of viral signal with endothelial damage, microthrombi, or lymphocytic infiltrates are combined with clinical data such as troponin kinetics and CMR findings A practical limitation of activity-based testing is RNA integrity, particularly in formalin-fixed, paraffin-embedded (FFPE) myocardial tissue, where formalin-induced crosslinking and fragmentation can markedly reduce reverse transcription quantitative PCR (RT-qPCR) sensitivity and increase false-negative results. Therefore, negative RNA findings—especially in low-copy DNA-positive samples or focal lesions—should be interpreted cautiously and ideally triangulated with localization methods (in situ hybridization, immunohistochemistry) and histopathological context [58,73,83].

EMB samples in which B19V DNA is present only at low copy number, without evidence of transcriptional activity or histopathological association with injury, have been proposed to be interpreted cautiously as indicating latent or incidental persistence rather than proven etiologic relevance [58,73,74]. This has been confirmed by studies, where reduction of B19V transcriptional activity rather than loss of DNA signal has been associated with clinical improvement after antiviral or immunomodulatory therapy [57]. In an endomyocardial biopsy (EMB)–guided paediatric protocol, myocardial inflammation prompted anti-inflammatory therapy (corticosteroids ± mycophenolate), whereas a positive myocardial viral PCR result triggered addition of antiviral treatment (e.g., interferon-β), and complete recovery was most frequently observed in episodes combining marked inflammation with positive myocardial PCR [84].

In veterinary medicine, the diagnostics reflect different clinical realities and tissue availability. In neonatal puppies that die suddenly or with signs of heart failure, necropsy with histopathological examination of the heart remains the primary diagnostic method. Multifocal or coalescing lymphoplasmacytic myocarditis with myofibre necrosis, intranuclear inclusion bodies in cardiomyocytes, and detection of CPV-2 antigen or DNA by immunohistochemistry or in situ hybridization in cardiomyocyte nuclei are regarded as strong evidence for causal involvement [13,60,75,76]. qPCR on myocardial tissue showed that CPV-2 DNA is significantly more frequent and present at higher loads in hearts with myocarditis or myocardial fibrosis than in age-matched controls [12].

In older puppies presenting with enteric CPV-2 infection, ante-mortem myocardial tissue is rarely available. Therefore, serial measurement of cardiac troponin I and echocardiographic techniques, including the Tei index and two-dimensional speckle-tracking echocardiography, have been used, with post-mortem correlation when possible [61,79,80,85]. These studies have reported transient systolic dysfunction and troponin release in a proportion of puppies with parvoviral enteritis, consistent with predominantly reversible myocardial injury in this age group rather than classical necrotizing myocarditis [80,85].

Substantial limitations remain in both human and veterinary diagnostics. Sampling error and the patchy, focal nature of lesions can produce false-negative EMB or necropsy findings even in clinically overt disease [12,74]. Assay design, contamination control, and validation are crucial for nucleic acid detection, particularly for small, stable DNA viruses such as parvoviruses, which can persist at low copy numbers in multiple tissues and may be inadvertently transferred between samples [83]. This risk further strengthens the rationale for the “evidence bundle” framework: isolated low-copy DNA positivity without transcriptional activity and without spatially concordant inflammation should be interpreted conservatively as possible incidental persistence or technical carryover rather than proof of etiologic involvement.

RNA-based assays are more specific for virus replication but are technically demanding because of RNA lability, especially in formalin-fixed, paraffin-embedded material. Finally, the absence of robust small-animal models that faithfully reproduce human B19V myocarditis has complicated validation and underscored that translational extrapolation from CPV-2 myocarditis in puppies to B19V-associated myocardial disease in humans must be based on converging virological, histological, and clinical evidence, rather than on DNA detection alone [31,86].

### 3.3. Clinical Consequences and Management in Different Host Groups

In human patients, B19V-associated myocarditis has been described across the entire age spectrum, with certain age-related patterns observed. In fetuses and newborns, intrauterine infection can result in severe anemia, hydrops fetalis, and fetal loss; when fetal myocarditis occurs, extensive cardiomyocyte damage and heart failure are observed [44]. In children and adolescents, B19V myocarditis often presents as acute heart failure, chest pain, or arrhythmias during or shortly after a febrile rash illness; clusters of pediatric myocarditis cases coinciding with B19V outbreaks have been reported [87,88]. In adults, presentation ranges from acute chest pain with preserved or mildly reduced systolic function to fulminant myocarditis requiring mechanical support to insidious progression to dilated cardiomyopathy [2,3].

In terms of management, no B19V-specific antiviral therapy has been established. Treatment is therefore largely supportive and follows general myocarditis guidelines, including arrhythmia management. In fulminant cases, temporary mechanical circulatory support and transplantation are indicated when necessary [1]. In patients with transcriptionally active B19V infection and chronic myocardial inflammation, interferon-β has been used and reported to reduce viral transcripts and improve cardiac function; however, the data remain limited and heterogeneous [89]. Intravenous immunoglobulin has been administered empirically in some cases of acute B19V myocarditis, particularly in children and immunocompromised hosts, with anecdotal improvement; however, controlled data are lacking [87]. Immunosuppressive therapy has been used in virus-negative inflammatory cardiomyopathy and in some B19V-positive cases with low viral loads and prominent autoimmune features, but concerns about promoting viral persistence have led to a cautious, case-by-case approach [1,2].

In dogs, clinical management of CPV-2 myocarditis depends strongly on age and presentation. For neonatal puppies with fulminant myocarditis, prognosis is generally poor; sudden death can occur before veterinary care is sought, and available interventions are largely supportive (oxygen supplementation, management of arrhythmias, fluid therapy) [13]. For surviving puppies with evidence of chronic myocardial fibrosis, standard therapies for canine dilated cardiomyopathy, including angiotensin-converting enzyme inhibitors, pimobendan, and diuretics as needed, can be instituted. Long-term monitoring for arrhythmias is also advisable [12]. In older puppies with enteric CPV-2 infection, the identification of elevated cardiac troponin I or impaired strain on echocardiography should prompt cautious fluid management, avoidance of cardiotoxic drugs, and possibly a cardiology consultation; nevertheless, many such changes appear reversible and may not progress to chronic cardiomyopathy [61,80].

Prevention strategies differ markedly between the two viruses. CPV-2 infection and its myocardial sequelae have been dramatically reduced through vaccination of not only puppies but also bitches, which ensures high titres of maternal antibodies during the vulnerable neonatal period of their offspring [90]. Although canine parvovirus type 2 (CPV-2) continues to diversify—largely through amino-acid substitutions in viral protein 2 (VP2)—with contemporary field strains clustering within CPV-2a, CPV-2b, and CPV-2c lineages (including “new” CPV-2a/2b sublineages), multiple studies have not demonstrated a consistent loss of vaccine protection against severe disease, supporting sustained vaccine effectiveness across these variants, including CPV-2c [91,92,93,94,95,96,97,98,99]. In contrast, there is currently no licensed vaccine against B19V, and infection is nearly universal by the time individuals reach adulthood. Therefore, prevention efforts have focused mainly on protecting seronegative pregnant women through hygiene and occupational measures during outbreaks [44]. The development of B19V vaccines based on VP1/VP2 virus-like particles has been explored but has not yet led to clinical availability [100]. The key viral effectors, best diagnostic evidence frameworks, functional/serum markers, persistence signals, typical clinical course, and prevention/therapy considerations discussed above are summarised in Table 3.

## 4. Synthesis and Practical Implications

When the biology and pathogenesis of B19V and CPV-2 are considered together, a continuum of parvoviral cardiac involvement can be appreciated. At one end of this continuum, CPV-2 in neonatal puppies exemplifies a scenario in which a small autonomous parvovirus directly infects and lyses proliferating cardiomyocytes, producing a fulminant necrotizing myocarditis that heals by fibrosis in survivors [12,13]. At the other end, B19V in adult humans represents an endotheliotropic *Erythroparvovirus* whose cardiac manifestations arise mainly from endothelial infection, microvascular dysfunction, and immune-mediated injury in a largely non-permissive myocardium, with viral persistence and host immune response jointly shaping the transition from acute inflammation to chronic cardiomyopathy [2,31,58].

Despite these differences, several shared principles emerge that can guide clinical reasoning. Both viruses demonstrate that parvoviral genomes can persist in cardiac tissue long after the acute infection has resolved, so that detection of DNA alone does not prove causality. Both encode NS1 proteins that trigger DNA damage and apoptosis, as well as small capsid-related phospholipases that are essential for cell entry and may contribute to membrane perturbation. Both elicit lymphocytic myocarditis and can culminate in fibrosis and ventricular remodelling. It is therefore suggested that clinicians and pathologists adopt a diagnostic approach that integrates virological, histopathological, and clinical data, rather than considering them in isolation.

For cardiologists evaluating patients with suspected B19V-associated myocarditis or dilated cardiomyopathy, it is recommended that EMB is accompanied by quantitative PCR for B19V DNA, assays for viral transcripts or proteins when feasible, and careful histological assessment for endothelial-centric lesions. Low-copy DNA in the absence of transcriptional activity or histological correlation should be interpreted cautiously and preferably reported as evidence of prior exposure or bystander persistence. When higher viral loads, viral mRNA, and lesion co-localisation are documented, a stronger etiologic link can be drawn, and, in such cases, consideration can be given to interferon-β or immunomodulatory therapies within the framework of clinical trials or specialist centres, although supportive heart failure therapy remains the cornerstone [1,55,89]

For veterinarians, particular attention should be paid to necropsies of young puppies with sudden death or unexplained heart failure, with systematic sampling of the heart for histopathology and CPV-2 detection, because such data continue to refine understanding of the epidemiology and long-term consequences of CPV-2 myocarditis [12,13]. In puppies with parvoviral enteritis, routine measurement of cardiac troponin I, along with speckle tracking echocardiography where available, may facilitate early recognition of myocardial involvement [61,80,101]. Continued emphasis on vaccination of dams and puppies remains crucial to prevent both enteric and cardiac manifestations of CPV-2 infection [90].

In conclusion, parvoviruses can cause myocardial damage through diverse mechanisms that depend on their tropism, host species, and age context. Understanding of these mechanisms, informed by comparative virology and underpinned by multi-layered diagnostic evidence, has been considered essential for accurate etiologic attribution and rational therapeutic decision-making in both human cardiology and veterinary practice.

## Figures and Tables

**Table 1 cimb-48-00052-t001:** Virion structure, receptors, and entry–CPV-2 vs. B19V.

Dimension	CPV-2	B19V
**ICTV placement**	Subfamily: *Parvovirinae* Genus: *Protoparvovirus* Species: *Carnivore protoparvovirus 1* [14,29]	Subfamily: *Parvovirinae* Genus: *Erythroparvovirus* Species: *Primate erythroparvovirus 1* [14,38]
**Genome/size**	Linear ssDNA genome ≈ 5.1 kb non-enveloped icosahedral capsid ≈ 25 nm [14,29]	Linear ssDNA genome ≈ 5.6 kb non-enveloped icosahedral capsid ≈ 22–24 nm [14,35,38]
**Capsid proteins**	T = 1 icosahedral capsid (60 subunits): VP2 major, VP1 minor; VP1u contains a conserved PLA2 motif [14,29,35]	T = 1 icosahedral capsid (60 subunits): VP2 major, VP1 minor; VP1u-externalised during entry; carries PLA2 plus receptor-interaction determinants [35,38]
**Primary receptor**	Canine TfR1—high-affinity binding and major determinant of host range restriction [29,30,41,44]	Globoside (P antigen) —essential attachment receptor; α5β1 integrin and additional co-receptors [38,48,49]
**Entry route**	TfR1-dependent, clathrin-mediated endocytosis; trafficking through acidified endosomes prior to capsid rearrangements [29,30,41,44]	P antigen-dependent, receptor-mediated endocytosis (supported by co-receptors such as α5β1 integrin); trafficking through acidified endosomes [38,48,49]
**Entry trigger**	Endosomal acidification triggers conformational rearrangements with VP1u exposure (at fivefold axes); capsid-tethered PLA2 promotes endosomal escape and enables nuclear delivery/import [35,39,44]	Endosomal acidification triggers VP1u exposure and PLA2 activity; efficient endosomal escape and nuclear delivery depend on the appropriate P-antigen/co-receptor context in permissive cells [29,35,39,48,49]

**Table 2 cimb-48-00052-t002:** Cardiac tropism and lesions–CPV-2 vs. B19V.

Dimension	CPV-2	B19V
**Main cardiac target**	Neonatal cardiomyocytes [1,60]	Microvascular endothelium; fetal cardiomyocytes in some cases [55,58,65]
**Age window**	Fetus/early neonate (late gestation–first weeks) [1,12,13]	Fetus (hydrops fetalis myocarditis) and all ages (endothelium) [2,3,58]
**Permissivity driver**	S phase dependence; high neonatal mitotic index [14,29,60]	Productive replication in erythroid progenitors; endothelial infection largely abortive [38]
**Dominant injury**	Lytic myocyte necrosis → myocarditis → replacement fibrosis [1,12,13]	Endothelial dysfunction/activation → microvascular injury → immune-mediated damage [3,55,58,65]
**Histologic hallmarks**	Myofiber loss; lymphoplasmacytic myocarditis; intranuclear inclusions; patchy replacement fibrosis [1,12,13]	Perivascular lymphocytic myocarditis; endothelial viral genomes; variable interstitial/replacement fibrosis [3,55,58,65]
**Clinical phenotype**	Peracute death or neonatal congestive heart failure; late scar-related dysfunction/arrhythmias [1,12]	Myocarditis phenotype (chest pain, acute heart failure, arrhythmias); subgroup → inflammatory DCM [2,3,58]

**Table 3 cimb-48-00052-t003:** Pathogenesis, diagnostics and prevention–CPV-2 vs. B19V.

Dimension	CPV-2	B19V
**Key viral effectors**	NS1 cytotoxicity; VP2–TfR1 drives tropism/host range [14,29,79]	NS1/11 kDa effects in endothelium; VP1u/VP2 + PLA2 (entry/escape) [20,56]
**Best diagnostic evidence**	Neonates: myocardial histology + IHC/ISH + PCR (heart) [1,12,13,26,80]	EMB bundle: high DNA load + RNA/antigen co-localisation + CMR/biomarkers [3,58]
**Functional/serum markers**	↑ cTnI; abnormal Tei index/strain (parvo enteritis pups) [85,90]	↑ troponin; CMR T1/T2/LGE (non-ischemic injury) [3,58]
**Persistence signal**	Viral DNA/RNA in myocarditis/fibrosis; later static scar [1,12,13]	Frequent latent DNA; outcome tracks active transcription (mRNA) [58]
**Typical course**	Fulminant neonatal myocarditis → fixed scars → late arrhythmias/HF [1,12,13]	Acute/subacute myocarditis; subset → chronic inflammatory cardiomyopathy/DCM [2,3,58]
**Prevention/therapy**	VP2 vaccines effective; myocarditis care mainly supportive [14,29,60]	No vaccine; standard HF/arrhythmia care ± selective immunomodulation; caution with DNA-only EMB [2,3,58]

## Data Availability

No new data were created or analyzed in this study. Data sharing is not applicable to this article.

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
