# Peer review of "Parvoviruses at the Heart: Endothelial Injury and Myocyte Lysis in Human B19V and Canine CPV-2 Infections"

_cimb, 2025, doi:10.3390/cimb48010052_

Round 1
Reviewer 1 Report
Comments and Suggestions for Authors
This is a well-structured, insightful, and timely comparative review that synthesizes complex virological and clinical information across human and veterinary medicine. There are several points for improvement as follows.
- Although the abstract gives a good summary, it could be improved by making the language more concise and the main conclusions clearer.
- Theintroduction provides a good context, but the transition from general viral myocarditis to specific parvoviruses (B19V and CPV-2) could be smoother.Consider adding a brief sentence to bridge this gap.
- Throughout the text, references to figures and tables are mentioned (e.g., "as shown in Figure X"), but the actual figures and tables are absent from the submitted manuscript.Ensure all referenced visuals are included in the final submission.
- There are minor errors in grammar and awkward expressions throughout the manuscript.A thorough proofread by a native English speaker or a professional editor is recommended.
Reviewer 2 Report
Comments and Suggestions for Authors
Summary:
The manuscript provides a thorough and well-researched comparative review of the cardiac manifestations associated with human parvovirus B19 (B19V) and canine parvovirus type 2 (CPV-2). Given the lack of robust animal models to reproduce human B19V myocarditis underscores how CPV-2 myocarditis translational extrapolation to related disease in humans must be based on converging virological, histological, and clinical evidence, rather than on diagnostic assessments alone. The topic is clinically relevant, especially given recent epidemiological shifts and renewed interest in parvoviral contributions to myocarditis. The authors demonstrate deep understanding and synthesis of the underlying virology, pathogenesis, diagnostic considerations, and clinical implications of both viruses. The manuscript integrates virology, host tropism, replication biology, immune injury, clinical presentation, and cross-species comparisons effectively. The authors provide insights into viral structural elements that directly contribute to viral tropism, subsequent injury and clinical implications, such as NS1-mediated injury, endothelial dysfunction, and CPV-2 neonatal cardiomyocyte susceptibility are well articulated. The cross-species synthesis is highly informative and beneficial for translational researchers and provides an intuitive progression from basic virology to diagnostic frameworks and clinical management. Inclusion of 2024–2025 B19V outbreak data and modern molecular insights adds significant value. The review is strong overall, but some minor improvements might enhance readability.
References are well organized and well cited with current and older published literature.
Minor Suggestions:
Given the manuscript’s density, visual aids would significantly enhance readability. There are several instances where the viral mechanistic insights are highly detailed in sections including Virion structure of B19V and CPV-2 [lines 121-160] and Replication cycles and cell tropism [ lines 162-216]. Describing these through figure illustrations would add value and help synthesize the intricate pathways, structural elements and virus-host cell interactions better. For instance, a comparative table (B19V vs. CPV-2) covering tropism, mechanisms of injury, diagnostic markers, and age susceptibility, and/or a schematic diagram showing pathways of endothelial injury vs. cardiomyocyte lysis. Such figures would help streamline complex content for readers. Tables with key comparative analysis between B19V and CPV-2 are provided within the supplementary material and the main manuscript might benefit from bringing them or a more high-level synthesis over to the main text.
While individual sections are comprehensive, the manuscript would benefit from more explicit connections between concepts. For instance: How differences in cellular tropism (endothelial vs. myocyte) shape diagnostic priorities. How viral persistence vs. active transcription influences clinical decision-making. The authors address much of this in the “Synthesis and practical implications” section but might benefit from more discussion.
B19V replication tropism [lines 181–199] and CPV-2 age-limited replication [lines 209–216] describe parallel themes might benefit from providing a link to diagnostic reasoning later [lines 286–337].
The evidence bundle approach used by the authors is well presented and explained but could be strengthened with highlighting additional limitations. RNA detection challenges are mentioned only briefly [lines 297–307] and highlighting the difficulty of RNA preservation in FFPE myocardial tissue, which directly impacts interpretation. Contamination concerns are noted [lines 328–336] and the authors may consider connecting these back to where the distinction between incidental DNA, active infection, viral transcriptional activity and associated inflammation is emphasized [lines 242–280].
Overall Recommendation: Minor Revision
The manuscript is scientifically sound, comprehensive, and timely. With improved synthesis, slight condensation of repetitive content, addition of visual aids, and minor editorial refinements, it will become a highly impactful review suitable for publication.
Reviewer 3 Report
Comments and Suggestions for Authors
Reviewer Comments
- Line 108- 111and 112-124. How did you predict all those features? Virion structures.
- Please include quantitative information on the number of protective virions released as a result of non-structural protein activity and clarify how these non-structural proteins contribute to virion assembly, release, or protection.
- Please specify which tissues are affected or injured by the capsid protein, including any known tissue-specific susceptibility or damage observed.
- Please comment on the genetic and phylogenetic proximity of the newly identified canine parvovirus type 2 to previously reported strains and discuss its relatedness to be known variants and any potential implications of this proximity.
